# Prevalence of Vitamin D Deficiency in a Large Newborn Cohort from Northern United States and Effect of Intrauterine Drug Exposure

**DOI:** 10.3390/nu12072085

**Published:** 2020-07-14

**Authors:** Neelakanta Kanike, Krupa Gowri Hospattankar, Amit Sharma, Sarah Worley, Sharon Groh-Wargo

**Affiliations:** 1Department of Pediatrics, Division of Neonatology, Case Western Reserve University, Metro Health Medical Center, 2500 MetroHealth Drive, Cleveland, OH 44109, USA; krupagowri.89@gmail.com (K.G.H.); sgrohwargo@metrohealth.org (S.G.-W.); 2Department of Pediatrics, Division of Neonatology, Memorial Hospital of Carbondale, IL 62901, USA; dramit55555@gmail.com; 3Department of Quantitative Health Sciences, Section of Biostatistics, Cleveland Clinic, Cleveland, OH 44195, USA; worleys@ccf.org

**Keywords:** vitamin D deficiency, newborn, preterm, maternal vitamin D deficiency, intrauterine drug exposure

## Abstract

Vitamin D is not only a vital element in bone health but is also a prohormone. Data regarding distribution of vitamin D status among preterm and term neonates in the United States are limited. There are no data on the effect of intrauterine drug exposure on vitamin D status. Our objective was to determine the distribution of vitamin D levels among preterm and term neonates and the effect of intrauterine illicit drug exposure. We did a retrospective chart review of neonates admitted from 2009 to 2016 to our neonatal intensive care unit with serum 25-hydroxycholecalciferol (25[OH]D) levels measured during the hospital stay. Of 1517 neonates, the median 25[OH]D level was 19 ng/mL with 31% deficient and 49% insufficient, even though 75% of mothers took prenatal vitamins. In pregnant women, 38% were vitamin-D-deficient and 44% were vitamin-D-insufficient. Four hundred seventy-one neonates had intrauterine drug exposure, with a median 25[OH]D level of 22.9 ng/mL versus 17.8 ng/mL in nonexposed neonates (*p* = 0.001). Despite maternal prenatal vitamin intake, neonates are at risk of vitamin D deficiency. Maternal illicit drug use was not related to lower 25[OH]D levels in neonates.

## 1. Introduction

Vitamin D is not only a vital element in bone health but is also a prohormone that has an important role in other body systems. Vitamin D deficiency in neonates has been linked to a higher risk of respiratory distress syndrome, food sensitivities, asthma, type I diabetes, autism, schizophrenia, and lower respiratory infections [1,2,3,4,5,6,7,8]. Serum 25-hydroxycholecalciferol (25[OH]D), the best estimator of body vitamin D stores, crosses the placenta through passive or facilitated transport according to a concentration gradient [9,10]. Deficiency of vitamin D is a worldwide health concern that affects more than one billion kids and adults globally [11]. Vitamin D status in the fetus and newborn infant is largely determined by maternal vitamin D status [9]. Deficiency of vitamin D in pregnant women may affect fetal growth, bone ossification, and tooth enamel formation [12]. Vitamin D deficiency during pregnancy increases the risk of adverse outcomes like pre-eclampsia, gestational diabetes mellitus, and low-birth-weight infants [13]. The main risk factor for vitamin D deficiency in neonates is maternal vitamin D deficiency [12]. In developed countries like the United States, vitamin D deficiency in neonates is emerging due to vitamin D deficiency in pregnant women [14,15]. Moreover, the data on the distribution of vitamin D status among the preterm population are limited.

In the last decade, illicit drug abuse in females of childbearing age has reached epidemic proportions in the USA [16,17,18]. It is known that people with substance use disorder typically suffer from nutrient deficiencies, and they have also been found to have altered hormonal metabolic regulators [19]. Drug abuse in pregnancy is related to numerous unfavorable outcomes for both mothers and infants, such as abruption of placenta, preterm delivery, cesarean section and prolonged hospitalization, and increased risk for maternal death [20]. For infants, maternal drug abuse in pregnancy is linked with prematurity, congenital anomalies, small size for gestational age, higher probability of pharmacological treatment for neonatal abstinence syndrome, prolonged hospitalization, and greater risk for neurodevelopmental issues [21]. Although the negative effects of illicit drug use on the fetus and newborn are well-documented, their effect on vitamin D levels is still unknown [22]. We hypothesized that neonates with intrauterine drug exposure have lower 25[OH]D levels at birth compared to the general population and that premature neonates have lower 25[OH]D levels than term neonates.

## 2. Materials and Methods

This was a single-institution retrospective chart review study done at Metro-Health Medical Center, affiliated with Case Western Reserve University, Cleveland, Ohio, United States of America (USA). The study cohort included neonates born between January 2009 and May 2016 with birth weight ≥1500 g and vitamin D levels measured during the hospital stay. Neonates with maternal history of disorder of the parathyroid glands, cholestasis of pregnancy, or maternal use of anti-retroviral drugs during pregnancy were excluded from the study.

Data regarding perinatal and postnatal characteristics were collected from the electronic health record system (Epic system corporation, US), including gender, gestational age, birth weight, race, 25[OH]D level, alkaline phosphatase level and type of intrauterine drug exposure. Serum 25[OH]D level was measured by liquid chromatography–tandem mass spectrometry, which is the gold standard [23]. Maternal data including vitamin D levels, body mass index (BMI), and usage of prenatal vitamins were collected. For analysis, we classified women and neonates into 4 groups based on 25[OH]D levels: (a) vitamin D deficiency (<15 ng/mL), (b) vitamin D insufficiency (15–30 ng/mL), (c) vitamin D sufficiency (31–100 ng/mL), and (d) vitamin D toxicity (>100 ng/mL). We used a cutoff of 25[OH]D levels less than 15 ng/mL as vitamin D deficiency based on the clinical practice in our institution. The same cutoffs were used for both infants and mothers, as experts argue that there is no reason to think the definition of vitamin D sufficiency varies by age [24]. We did a pilot study at our institution to screen for vitamin D deficiency in 2008 and found that out of 183 infants, 51% had vitamin D insufficiency and 19% had deficiency. Based on recommendations from the Endocrine Society Clinical Practice Guidelines, we developed a routine practice of measuring vitamin D levels at 24 to 48 h of life in our neonatal intensive care unit to identify infants with vitamin D deficiency and supplement them with 2000 IU ergocalciferol daily until the levels improved to a normal range (25[OH]D level >30 ng/mL) [25]. Metro-Health Medical Center’s institutional review board approved the study.

## 3. Statistical Methods

Data were presented using counts and percentages for categorical variables and medians, ranges, and quartiles for continuous variables. The vitamin D levels of subgroups were compared using chi-squared or Fisher’s exact tests for categorical characteristics and Wilcoxon rank-sum tests for continuous and ordinal characteristics. The associations between vitamin D levels and other continuous clinical characteristics were assessed using non-parametric Spearman rank correlation coefficients. Linear regression models were used to compare drug-exposed to non-drug-exposed neonates and preterm to term newborn 25[OH]D levels at birth while adjusting for other maternal and infant characteristics. A subgroup analysis of infants was performed to assess the association between maternal BMI and infant vitamin D levels. All analyses were performed on a complete-case basis. All tests were two-tailed and used a significance level of 0.05. For all analyses, SAS 9.4 software (SAS Institute, Cary, NC) was used.

## 4. Results

Of a total of 1517 neonates, 45% were term and 55% preterm. Table 1 shows descriptive data for our study cohort. The median gestational age was 36 weeks (30–42 weeks). The median birth weight was 2560 g (1500–4561 g), and males constituted 55% of the study population. Caucasian and African American (AA) neonates constituted 46% and 41%, respectively. Median 25[OH]D was 19 ng/mL (range, 3–223 ng/mL), with 31% being deficient and 49% being insufficient, even though 75% of mothers were on prenatal vitamins. In the vitamin-D-deficient group, 63% were AA and 27% were Caucasian (Table 2). Vitamin D levels were similar among preterm and term infants in our study cohort with a median 25[OH]D of 18.5 ng/mL and 18.9 ng/mL, respectively.

Intrauterine drug exposure was found in 471 neonates (31%). The most common illicit drugs were opioids and marijuana. Contrary to our hypothesis, the infants with intrauterine drug exposure had higher a median 25[OH]D level compared to nonexposed infants (22.9 ng/mL vs. 17.8 ng/mL, *p* = 0.001). On subgroup analysis, the levels were significantly higher in exposed infants among term and late-preterm groups (25.5 ng/mL vs. 16.8 ng/mL, *p* < 0.001, in term group; 21 ng/mL vs. 18.5 ng/mL, *p* = 0.01, in the late-preterm group) (Table 3).

Infant 25[OH]D levels were positively associated with mother 25[OH]D levels (0.64; 95% CI: 0.42–0.86; *p* < 0.001) and negatively correlated with alkaline phosphatase (−0.13, 95% CI: −0.22 to −0.04; *p* < 0.003). The 25[OH]D values were not found to have a correlation with birth weight, gestational age, or maternal BMI (Table 4). During pregnancy, 38% women were vitamin D deficient and 44% insufficient, though our sample size was small (n = 52).

## 5. Discussion

Deficiency of vitamin D is the most common cause of rickets and is also known to increase the risk of respiratory distress syndrome, food sensitivities, asthma, type I diabetes, autism, schizophrenia, and lower respiratory infections [1,2,5,7,26]. Absorption of dietary calcium and phosphorus is prevented by vitamin D deficiency. Vitamin D status in newborns is entirely dependent on maternal supply during pregnancy [10]. As expected, low maternal vitamin D status during pregnancy is a major risk factor for rickets in infants [12,27,28]. Rickets in children is caused by severe, chronic vitamin D deficiency with apparent skeletal abnormalities [29,30], but neonates with vitamin D insufficiency have no overt skeletal or calcium metabolism defects [24]. Rickets was a global disease in the early twentieth century but nearly disappeared in developed countries after its causal pathway was understood and fortification of milk with the hormone vitamin D was introduced at the population level [31]. Surprisingly, rickets is re-emerging per recent evidence [24,26], and vitamin D deficiency is prevalent in both developed and developing countries [26,32,33,34,35,36,37,38].

In our cohort of neonates born in the Cleveland area (latitude 41 °N), we report a remarkably high prevalence of vitamin D deficiency and insufficiency at birth of 31% and 49%, respectively. We noticed poor vitamin D stores especially in neonates, even though 75% of mothers reported regular multivitamin intake during pregnancy. In our cohort, vitamin D deficiency was more prevalent in African American (AA) neonates (63%) than Caucasian (27%) neonates. Our results are consistent with the study on 400 mother–infant dyads from Pittsburgh by Bodnar et al., who showed that nearly 50% of AA neonates and 10% of white neonates had serum 25[OH]D levels less than 15 ng/mL at birth despite adequate compliance with at least 400 IU vitamin D supplementation daily by 90% of their mothers [14]. A prolonged winter season with limited sun exposure in Cleveland might be contributing to the vitamin D deficiency.

Comparably, our results are also similar to a 1983 cross-sectional study by Hollis et al. on 10 AA and 12 Caucasian mothers who resided in the Cleveland area: serum 25[OH]D levels at term were lower in AA mothers (18.2 ± 12.1 ng/mL) than in white mothers (27.4 ± 8.7 ng/mL, *p* < 0.05) and were lower in AA neonates (9.5 ± 6.7 ng/mL) than in white neonates (14.9 ± 5.4 ng/mL, *p* < 0.05) [10]. We did not notice any substantial improvement in the vitamin D status among neonates born to AA women in the last 3 decades in the Cleveland area. A cross-sectional study by Merewood et al. from 2005 to 2007 in an urban Massachusetts area reported that 35.8% of mothers and 58% of neonates had vitamin D deficiency (25[OH]D <20 ng/mL); 23.1% of the mothers and 38.0% of the neonates had severe deficiency (25[OH]D <15 ng/mL). Risk factors for neonatal vitamin D deficiency were maternal vitamin D deficiency (adjusted odds ratio [aOR]: 5.28), winter birth (aOR: 3.86), AA race (aOR: 3.36), and maternal BMI of 35 (aOR: 2.78) [39]. Contrary to their results, we did not find any significant correlation between infant vitamin D levels and maternal BMI in our study.

The strong association between maternal and infant 25[OH]D levels reported in our study offers further confirmation that newborn 25[OH]D levels are dependent on maternal serum 25[OH]D levels as reported in previous studies [10,39,40]. Pregnant women should take 600 IU vitamin D supplements daily, preferably as a combined preparation with other micronutrients such as iron and folic acid per global consensus recommendations on prevention and management of nutritional rickets [41]. The Endocrine Society Clinical Practice Guidelines also recommend at least 600 IU of vitamin D supplementation daily in pregnant and lactating women. They also acknowledge that 1500–2000 IU/day of vitamin D may be necessary to keep 25[OH]D levels >30 ng/mL [25]. However, average prenatal supplements contain only 400 IU of vitamin D [41].

Rostami et al. evaluated the usefulness of a prenatal screening program in improving vitamin D status through pregnancy. The outcome of their screening program was the prevention of pregnancy complications. They witnessed greater than a 25-fold increase in the number of mothers who were able to achieve a vitamin D level greater than 20 ng/mL once they were screened for their vitamin D levels and provided supplementation during pregnancy compared with mothers who were not screened and so were not instructed to take additional vitamin D supplements. They reported a significant decrease in pregnancy complications for mothers in the screened group: 60%, 50%, and 40% drops in pre-eclampsia, gestational diabetes, and preterm delivery, respectively [42].

Presently, the data on the distribution of vitamin D levels in preterm neonates are limited. Few studies have reported vitamin D levels in neonates at birth, with sample sizes ranging from 8 to 223 neonates [1,43,44,45,46,47] and mean 25[OH]D levels ranging from ~6.5 ng/mL among preterm neonates in the United Arab Emirates [46] to 26.8 ng/mL among preterm neonates in Canada [48]. Our cohort had 832 preterm infants, the largest group reported to date. In contrast to the Boston group, in which preterm neonates had lower vitamin D levels, the Canada study and our study did not show any significant change in vitamin D status between term and preterm neonates [47,48].

Information about the effect of illicit drug use in pregnancy on the vitamin D levels in neonates is lacking in the literature. To date, this is the first study to look into the vitamin D status of neonates born to mothers with illicit drug use. The literature shows that people with drug use disorder typically suffer from nutrient deficiencies [19]. We expected lower vitamin D levels in the neonates born to women with illicit drug use due to poor nutrition during pregnancy, but the vitamin D levels of neonates with intrauterine drug exposure in our study cohort were not lower compared to nonexposed neonates; on the contrary, the levels were higher. The reason for this difference is not clear. Illicit drugs like marijuana, opiates, and cocaine may influence maternal–fetal transference of vitamin D. It is a novel relationship that needs further exploration.

This is the one of the largest studies on this topic in the United States with a cohort of 1517 neonates. Most of the similar studies used higher threshold cutoffs to define vitamin D deficiency (<20 ng/mL) compared to our study [15,39,49]. We used a level of 25[OH]D < 15 ng/mL for vitamin D deficiency based on the clinical practice in our institution. If we had used levels less than 20 ng/mL, the proportion of neonates in the vitamin-D-deficient group would have been even higher in our cohort.

We recognize some limitations in our study, including the small sample size of pregnant women with measured vitamin D levels, only 52 out of 1517, as the obstetrics team does not routinely screen for vitamin D status in women during pregnancy at our institution. As the number of mothers with their vitamin D levels checked was low, the correlation between newborn and maternal levels should be interpreted cautiously, although they are in the same direction as previously published works [50]. Our study population reflects the inner-city population in the Cleveland area of the Midwestern USA. The results cannot be generalized to the whole US population, although our study population is similar to most inner-city populations in major cities. Another limitation is lack of data on the duration and amount of illicit drug use, smoking and alcohol usage history, and nutritional intake of vitamin-D-fortified foods during pregnancy and the birth season. Future large, multicenter, prospective studies are required to know the prevalence of vitamin D deficiency in developed countries like the USA and to determine factors that are related to vitamin D deficiency.

## 6. Conclusions

Our study results suggest that neonates born in the Northern US are at high risk of vitamin D deficiency, even when pregnant women are compliant with recommended prenatal vitamin intake. Current prenatal multivitamins do not contain sufficient vitamin D to prevent deficiency. Higher-dose vitamin D supplementation may be required to improve maternal and neonatal vitamin D status. Future studies are necessary to determine the minimum dose of vitamin D supplementation required during pregnancy to accomplish vitamin D sufficiency. Maternal illicit drug use was associated with significantly higher neonatal vitamin D levels. Further large studies are needed to show the effect of illicit drug use in pregnancy on neonatal vitamin D status. It is time to reconsider our approach to ensure vitamin D sufficiency in pregnant women and their infants.

## Figures and Tables

**Table 1 nutrients-12-02085-t001:** Maternal and infant descriptive data.

Variables	Number (%) Total: 1517
Gestational Age, Median (Min, Max)	36(30,42)
Gestational age	
Term (≥37 weeks)	685(45)
Preterm (<37 weeks)	832(55)
Groups Based on Birth Gestation	
Early Preterm (<34 weeks)	335(22)
Late Preterm (34–36 weeks)	497(33)
Term (≥37 weeks)	685(45)
Gender	
Male	830(55)
Ethnicity	
Caucasian	692(46)
African American	626(41)
Birth Weight, Median (Min, Max)	2559(1500,4561)
Size	
SGA—Small for Gestational Age	185(12)
AGA—Appropriate for Gestational Age	1191(79)
LGA—Large for Gestational Age	97(6)
Vitamin D Level, Median (Min, Max)	19(3,223)
Vitamin D Level, 25[OH]D	
Deficiency (<15 ng/mL)	476(31)
Insufficiency (15–30 ng/mL)	747(49)
Sufficiency (31–100 ng/mL)	288(19)
Toxicity (>100 ng/mL)	4(0.3)
Alkaline Phosphatase Level Available	489(32)
Alkaline Phosphatase Level, Median (Min, Max)	175(22,697)
Maternal Drug Exposure	471(31)
Opiate	210(14)
Marijuana	156(10)
Cocaine	65(4)
Benzodiazepine	10(0.7)
Polydrug Use	133(28)
Prenatal Vitamin	1133(75)
Mothers with Vitamin D Level	52(4)
Mother’s 25[OH]D level, Median (Min, Max)	18(4,64)
Mother’s Vitamin D Level, 25[OH]D	
Deficiency (<15 ng/mL)	20(38)
Insufficiency (15–30 ng/mL)	23(44)
Sufficiency (31–100 ng/mL)	9(17)

Data shown as number (%) unless otherwise stated.

**Table 2 nutrients-12-02085-t002:** Descriptive statistics by vitamin D status.

Variable, N (%)	Deficiency (N = 476)	Insufficiency (N = 747)	Sufficiency (N = 288)	Toxicity (N = 4)
Gender: Male	252 (53)	421 (51)	154 (19)	2 (0.2)
Ethnicity:				
Caucasian	130 (27)	366 (53)	192 (28)	3 (0.4)
African American	302 (63)	258 (41)	64 (10)	1 (0.2)
Birth Gestation:				
Early Preterm (<34 w)	107 (32)	178 (53)	50 (15)	0 (0)
Late Preterm (34–36 w)	155 (31)	251 (51)	89 (18)	1 (0.2)
Term (≥37 w)	214 (31)	318 (46)	149 (22)	3 (0.4)
Mother’s BMI, Median (Q1, Q3)	30 (24,39)	27 (23,33)	28 (25,33)	28 (28,28)
Prenatal Vitamin	361 (76)	540 (72)	228 (79)	3 (75)
Mother’s Vitamin D Level Median (Q1, Q3)	9 (9,15)	22 (15,29)	25 (20,31)	----
Mothers Drug Exposure	113 (24)	210 (28)	143 (50)	3 (75)
Opiate	47 (42)	99 (47)	62 (43)	2 (66)
Marijuana	60 (53)	77 (37)	18 (13)	0 (0)
Cocaine	16 (14)	37 (18)	12 (8)	0 (0)
Benzodiazepine	3 (3)	5 (2)	2 (1)	0 (0)

BMI: body mass index.

**Table 3 nutrients-12-02085-t003:** Neonatal vitamin D level by maternal drug exposure within age groups.

Gestational Age Group	Intrauterine Illicit Drug Exposure	Number of Subjects	Vitamin D Level (ng/mL) Median (P25, P75)	*p* Value
All Neonates	Yes	469	22.9 (15, 33.2)	<0.001
	No	1045	17.8 (13, 24.7)	
Term Neonates (≥37 weeks)	Yes	234	25.5 (15.9, 35)	<0.001
	No	449	16.8 (12.4, 24)	
Late Preterm (34–36 weeks)	Yes	151	21 (14,33)	0.010
	No	345	18.5 (13.7, 24.8)	
Early Preterm (<34 weeks)	Yes	84	19.7 (14.1, 29)	0.22
	No	251	18.6 (13.3, 25.4)	

**Table 4 nutrients-12-02085-t004:** Association of continuous characteristics with infant vitamin D levels. *

Characteristics	N	Spearman Correlation Coefficient	95% CI	*p* Value
Neonate Alkaline Phosphatase	489	−0.13	(−0.22,−0.04)	0.003
Birth Weight	1515	0	(−0.05,0.05)	0.98
Gestational Age	1515	0.02	(−0.04,0.07)	0.55
Maternal Vitamin D Level	52	0.64	(0.42,0.86)	<0.001
Maternal BMI	232	−0.03	(−0.16,0.10)	0.63

***** Correlation coefficients >0 indicate a positive association between 25[OH]D and the characteristic (higher levels of one are correlated with higher levels of the other), and correlation coefficients <0 indicate a negative association between 25[OH]D and the characteristic (higher levels of one are associated with lower levels of the other); CI: confidence interval; BMI: body mass index.

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
