# Peer review of "Prevalence of Vitamin D Deficiency in a Large Newborn Cohort from Northern United States and Effect of Intrauterine Drug Exposure"

_nutrients, 2020, doi:10.3390/nu12072085_

Round 1

Reviewer 1 Report

Observational study on a large cohort of newborns in Northern US. The authors have tried to correlate findings with vi D levels in mothers and with the use of illicit drugs during pregnancy.  The have stratified the population according to be a term or a preterm baby.

There is a clear research hypothesis and the design is appropriate to answer that question.

Comments

  1. This subject has been studied previously in different scenarios. The authors should provide more references on previous similar studies. As a sample:
    1. Wang C, Gao J, Liu N, Yu S, Qiu L, Wang D. Maternal factors associated with neonatal vitamin D deficiency. J Pediatr Endocrinol Metab. 2019;32(2):167-172. doi:10.1515/jpem-2018-0422
    2. Alok Sachan, Renu Gupta, Vinita Das, Anjoo Agarwal, Pradeep K Awasthi, Vijayalakshmi Bhatia, High prevalence of vitamin D deficiency among pregnant women and their newborns in northern India, The American Journal of Clinical Nutrition, Volume 81, Issue 5, May 2005, Pages 1060–1064, https://doi.org/10.1093/ajcn/81.5.1060
  2. Material and methods
    1. It would have been interesting to gather data from the tome of delivery as mothers who deliver in winter or early spring use to have lower vit D levels.
    2. Please, provide some information on calcium status in newborns and, if possible, Alk Phos (not provided)
    3. Please, provide mor information regarding the reason the ask for a vit D level in those newborns as it is not a routine lab: seizures? Tremors?
    4. Were there mothers with multiuse of drugs? If so, provide.
    5. As tobacco exposure during pregnancy may influence vit D levels (Banihosseini SZ, Baheiraei A, Shirzad N, Heshmat R, Mohsenifar A. The effect of cigarette smoke exposure on vitamin D level and biochemical parameters of mothers and neonates. J Diabetes Metab Disord. 2013;12(1):19. Published 2013 May 11. doi:10.1186/2251-6581-12-19), provide information on tobacco exposure.
    6. Provide information on the method to measure vit D levels
  3. Results
    1. Table 1: the sum of mothers with Vit d levels was 52 not 56. Clarify
    2. As in the group with deficiency mothers have a higher BMI (30) a a lower use of  prenatal vitamin supplements, results need to be reassess according to these variables.
    3. Table 2. As SSRI exposure in all groups is 0, the file can be deleted.
    4. In table 2. The % of use of each type of drug should be presented within the same group not as % of the file.
    5. As the number of mothers with vit D checked is so low, the correlation between levels on the newborn and maternal levels should be interpreted cautiously, although they are in the same direction of previous published works.

Author Response

Author’s response to reviewers’ comments

 Observational study on a large cohort of newborns in Northern US. The authors have tried to correlate findings with vitamin D levels in mothers and with the use of illicit drugs during pregnancy.  The have stratified the population according to be a term or a preterm baby.

There is a clear research hypothesis and the design is appropriate to answer that question.

Comments

  1. This subject has been studied previously in different scenarios. The authors should provide more references on previous similar studies. As a sample:

- Wang C, Gao J, Liu N, Yu S, Qiu L, Wang D. Maternal factors associated with neonatal vitamin D deficiency. J Pediatr Endocrinol Metab. 2019;32(2):167-172. doi:10.1515/jpem-2018-0422

-Alok Sachan, Renu Gupta, Vinita Das, Anjoo Agarwal, Pradeep K Awasthi, Vijayalakshmi Bhatia, High prevalence of vitamin D deficiency among pregnant women and their newborns in northern India, The American Journal of Clinical Nutrition, Volume 81, Issue 5, May 2005, Pages 1060–1064, https://doi.org/10.1093/ajcn/81.5.1060

Author Response: Thank you for suggesting these studies. We have now cited these studies in the revised manuscript in the discussion section (line 132, references 31 & 32).

Material and methods:

  1. It would have been interesting to gather data from the time of delivery as mothers who deliver in winter or early spring use to have lower vitamin D levels.

Author Response: This is an important observation and we agree with the reviewer comment that infants born in winter or early spring tend to have lower vitamin D levels. But unfortunately, we did not collect the date of the birth of the infants in the study group as it was a patient identifier as per our institutional review board. We included this as a limitation in the discussion section of revised manuscript (pages 200-202). 

  1. Please, provide some information on calcium status in newborns and, if possible, Alkaline Phosphatase (not provided).

Author Response: We agree with the reviewer regarding need for this data. Alkaline Phosphatase data was available in 489 infants (32%). We updated table 1 with neonatal alkaline phosphatase levels. Unfortunately, data on the calcium and phosphorus was not available in most of the late preterm and term infants.

  1. Please, provide more information regarding the reason the ask for a vit D level in those newborns as it is not a routine lab: seizures? Tremors?

Author Response: We did a pilot study at our institution to screen for vitamin D deficiency in 2008 and found that out of 183 infants, 51% of infants had vitamin D insufficiency and 19% had deficiency.  Based on the recommendations from Endocrine society clinical practice guidelines, we developed a practice of measuring Vitamin D levels at 24 to 48 hours of life routinely in our neonatal intensive care unit to identify infants with vitamin D deficiency and supplement them infants with 2000 IU Ergocalciferol till the levels improve to normal range (25[OH]D level >30 ng/mL).

  1. Were there mothers with multiuse of drugs? If so, provide.

Author Response: There were 133(28%) mothers with multidrug use. Commonly used drugs are marijuana and opiates or marijuana and cocaine. Table 1 has been updated with data on poly-drug use.

  1. As tobacco exposure during pregnancy may influence vitamin D levels (Banihosseini SZ, Baheiraei A, Shirzad N, Heshmat R, Mohsenifar A. The effect of cigarette smoke exposure on vitamin D level and biochemical parameters of mothers and neonates. J Diabetes Metab Disord. 2013;12(1):19. Published 2013 May 11. doi:10.1186/2251-6581-12-19), provide information on tobacco exposure.

Author Response: We agree with the reviewer, tobacco exposure during pregnancy may affect bone metabolism and leads to diminished maternal and neonatal bone mass density. We did not include maternal tobacco exposure in our study design as the study hypothesis was mainly about intrauterine drug exposure and prevalence of vitamin D deficiency.  

  1. Provide information on the method to measure vitamin D levels

Author Response: We appreciate the reviewer for this suggestion. We have added the information regarding the method to measure vitamin D level in the revised manuscript (lines 62 & 63) along with reference (18).

Results

  1. Table 1: the sum of mothers with Vitamin d levels was 52 not 56. Clarify

Author Response: We thank the reviewer for finding this error. As the reviewer mentioned, the sum is actually 52.  We have corrected the number in the revised manuscript (line 112 & 203) and table 1.

  1. As in the group with deficiency mothers have a higher BMI (30), a lower use of  prenatal vitamin supplements, results need to be reassess according to these variables.

Author Response: We acknowledge the reviewer for this observation. In the group with vitamin D deficiency, prenatal vitamin usage was 76%; in the group with vitamin D insufficiency prenatal vitamin usage was 72%; in the group with vitamin D sufficiency prenatal vitamin usage was 79% and in the group with vitamin D toxicity prenatal vitamin usage 75%. We have updated the percentages in table 2.

We agree that higher BMI may have an effect on vitamin D level. But our study analysis showed that vitamin D level have a negative correction with maternal BMI, but these results were not statistically significant.

  1. Table 2. As SSRI exposure in all groups is 0, the file can be deleted.

Author Response: As advised, we have deleted SSRI row from table 2 in the revised manuscript.

  1. In table 2. The % of use of each type of drug should be presented within the same group not as % of the file.

Author Response: We thank the reviewer for the suggestion. We have updated the percentages of use of each type of drug in table 2 in the revised manuscript.

  1. As the number of mothers with vitamin D checked is so low, the correlation between levels on the newborn and maternal levels should be interpreted cautiously, although they are in the same direction of previous published works.

Author Response: We agree with the reviewer that the sample size for maternal vitamin D levels is small.  The Obstetrics team at our institution do not routinely screen mothers for vitamin D deficiency.  We mentioned this as a limitation in the discussion section of the manuscript. As the reviewer mentioned, the results are in the same direction of previous published works.

Reviewer 2 Report

The authors of “Prevalence of Vitamin D Deficiency in a large Newborn Cohort from Northern Unites States and effect of Intrauterine Drug exposure” showed that despite maternal prenatal vitamin intake, neonates are at risk of vitamin-D deficiency. Maternal illicit drug use was not associated with lower vitamin-D levels in neonates. Few questions:

  1. Do the authors have information on when the illicit drug was taken in the first trimester, during the whole pregnancy?
  2. Are there data on the amount of illicit drugs and is there a correlation with vitamin D levels of newborn?
  3. It would be useful to know if there are data on the nutrition of mothers with foods fortified with vitamin D
  4. It would be helpful to know how long mothers spent each day in the sun during pregnancy
  5. Did mothers take other medicines during pregnancy ?. If so, which ones? Is there any relationship with vitamin D levels

Author Response

The authors of “Prevalence of Vitamin D Deficiency in a large Newborn Cohort from Northern Unites States and effect of Intrauterine Drug exposure” showed that despite maternal prenatal vitamin intake, neonates are at risk of vitamin-D deficiency. Maternal illicit drug use was not associated with lower vitamin-D levels in neonates.

Few questions:

  1. Do the authors have information on when the illicit drug was taken in the first trimester, during the whole pregnancy?

Author Response: We do not have information when the illicit drugs were exactly taken during pregnancy and how long. We collected the illicit drug data based on maternal urine drug screens at the time of delivery which represents drug usage in the last trimester.  We added this this as a limitation in the discussion of revised manuscript (pages 206-208)

  1. Are there data on the amount of illicit drugs and is there a correlation with vitamin D levels of newborn?

Author Response: We agree with the reviewer that amount of illicit drugs may have an impact of vitamin D levels. But we do not have data on the amount of illicit drugs taken during pregnancy. We collected the illicit drug data based on maternal urine drug screens at the time of delivery. Urine drug screens used, provide only qualitative data regarding the type of drugs only. We included this as a limitation in the discussion section of revised manuscript (pages 206-208)  

  1. It would be useful to know if there are data on the nutrition of mothers with foods fortified with vitamin D

Author Response: We agree with the reviewer suggestion. As mother being only sole source of vitamin D for fetus, her nutrition has significant impact of fetus nutrient levels. Unfortunately, being a retrospective study, we did not have the data on the nutrition of mothers with vitamin D fortified foods. We updated this as a limitation in the revised manuscript (pages 206-208)

  1. It would be helpful to know how long mothers spent each day in the sun during pregnancy

Author Response: We agree with the reviewer suggestion. As this is subjective, it will be difficult to collect this information with high potential for recall bias. Being a retrospective study we did not have the data on duration of sun exposure during pregnancy.

  1. Did mothers take other medicines during pregnancy? If so, which ones? Is there any relationship with vitamin D levels?

Author Response: We excluded mothers from the study who has history of parathyroid disorders, cholestasis of pregnancy or any use of antiretroviral drugs during pregnancy as these conditions may have effect on vitamin D levels. In the study group, we did not collect medications data other than prenatal vitamins.

Round 2

Reviewer 1 Report

The authors have answered most of my previous questions. In my opinion the paper can be accepted for publication